# A Bright Future for Fluorescence Imaging of Fungi in Living Hosts

**DOI:** 10.3390/jof5020029

**Published:** 2019-04-03

**Authors:** Ambre F. Chapuis, Elizabeth R. Ballou, Donna M. MacCallum

**Affiliations:** 1MRC Centre for Medical Mycology at the University of Aberdeen, Institute of Medical Sciences, University of Aberdeen, Foresterhill, Aberdeen AB25 2ZD, UK; r01afc16@abdn.ac.uk; 2Institute of Microbiology and Infection, School of Biosciences, University of Birmingham, Birmingham B15 2TT, UK; E.R.Ballou@bham.ac.uk

**Keywords:** fluorescent reporters, bioluminescence, iRFP, live imaging, infection models, 3Rs

## Abstract

Traditional *in vivo* investigation of fungal infection and new antifungal therapies in mouse models is usually carried out using post mortem methodologies. However, biomedical imaging techniques focusing on non-invasive techniques using bioluminescent and fluorescent proteins have become valuable tools. These new techniques address ethical concerns as they allow reduction in the number of animals required to evaluate new antifungal therapies. They also allow better understanding of the growth and spread of the pathogen during infection. In this review, we concentrate on imaging technologies using different fungal reporter proteins. We discuss the advantages and limitations of these different reporters and compare the efficacy of bioluminescent and fluorescent proteins for fungal research.

## 1. Fungi and Infection

Infections by fungi, e.g., *Candida*, *Aspergillus* and *Cryptococcus* species, are increasing threats to human health [1]. Some fungal pathogens colonise particular sites in the human body and, most of the time, do not cause harm to healthy individuals. However, these pathogens are significant threats to immunocompromised patients when the immune system is no longer able to control colonisation and infection develops. Life-threatening diseases, such as candidiasis or aspergillosis, mainly affect severely immunocompromised hosts, and cryptococcosis occurs mainly in HIV/AIDS patients [1,2,3]. As eukaryotic pathogens causing a range of infections, pathogenic fungi present specific challenges to human health and it is essential that we develop a better understanding of their pathogenicity and interactions with the host.

Current antifungals to treat fungal infections are limited to four classes: polyenes, azoles, echinocandins and 5-fluorocytosine (5-FC) [4,5], with amphotericin B (a polyene), fluconazole (an azole) and 5-FC featuring on the WHO Essential Medicine List [6]. However, these drugs are not fully effective in severe infections and are toxic to the patient [7,8]. The availability of amphotericin B and 5-FC in certain countries in Africa with high burdens of fungal infection and the local high cost of other antifungals, such as fluconazole, is also a global health issue [9,10]. Another issue in antifungal treatment is the difference in efficacy of a drug in different fungal infections; e.g., echinocandin drugs, such as caspofungin or micafungin, are effective against candidiasis [11,12]; however, they have no impact on *Cryptococcus* infections [13]. Furthermore, the increasing incidence of drug resistance, such as resistance to azoles and echinocandins in *Candida* and *Aspergillus* species, has become a severe clinical challenge [14,15,16]. In addition, the emergence of new species, such as *Candida auris,* which are resistant to most of the currently available antifungals, are a growing health threat [17]. Due to the limited availability of treatment choices, novel therapies are needed to improve patient outcome. 

Rodent, mainly mouse, models of fungal infection have been central to efforts to evaluate novel antifungal drugs. *In vivo* experiments allow essential understanding of pharmacokinetics and drug susceptibility, although emerging *in vitro* model systems enable increasingly complex investigations of interactions between host cells and fungi [18]. However, *in vitro* studies remain inherently limited in their capacity to reproduce the complexity of living organisms. In fact, scientific breakthroughs for some fungal pathogens have explicitly required animal models, e.g., *Pneumocystis jirovecii,* an obligate biotroph of the lung [19]. Even among readily cultured fungal species, host-specific phenomena, e.g., *Cryptococcus neoformans* titan cells [20], were only observed in hosts until the recent development of novel *in vitro* models [21,22,23]. Likewise, discovery of the effect of carbon source on *Candida albicans* virulence was first observed in murine infection models [24].

In addition to basic fungal biology insights that can be gained by studying these pathogens in the context of the host, animal models are central to efforts to develop new chemotherapeutics. Animal studies are mandatory for safety reasons, as some compounds, such as nystatin and amphotericin B, possess outstanding activity *in vitro* but show toxicity once administered to a host [25]. A suitable animal model is, therefore, essential to recognise the efficacy of any therapeutic agent, to optimise its mode of delivery, and to assess drug–host interactions during treatment of fungal infections [26]. Traditional *in vivo* investigations of drug efficacy in mice usually involve post-mortem end point methodologies, such as determining organ fungal burdens or measuring mouse survival during treatment [26]. These methods have been the gold standard but can require large numbers of animals to demonstrate drug efficacy. In addition, cryptic infection sites may be missed when these models are used [27]. 

## 2. Novel Non-Invasive Techniques

Recently, non-invasive imaging techniques have been developed to monitor infection progression, or resolution, allowing information to be obtained from individual animals over time [28,29]. In comparison with traditional protocols to measure drug efficacy, where groups of animals are sampled at several time points during treatment and the average response is measured, non-invasive imaging allows the response of each individual animal to be measured, reducing biological variation, and thus reducing the number of animals required per group to evaluate any new antifungal drug. These novel *in vivo* imaging techniques take advantage of bioluminescent or fluorescent proteins [30,31]. 

Bioluminescence refers to the light emitted from living organisms, which results from the oxidation of organic substrates mediated by luciferases [31]. Bioluminescent imaging has the advantage of being non-invasive, non-toxic for animals, and highly sensitive, with excellent signal-to-noise ratios [27]. However, a major drawback is the need to administer a substrate, such as luciferin. Attempts to investigate systemic candidiasis using bioluminescent reporters have failed due to poor penetration of the substrate in the mouse [32]. 

Fluorescent reporters emit light without injection of a substrate prior to imaging. Conventional green fluorescent proteins (GFPs) are not suitable for mammalian deep tissue imaging due to haemoglobin, water and skin melanin absorbing at the GFP emission wavelength, which leads to a high background [30,32,33,34]. However, fluorescent proteins in the near-infrared spectrum (650–900nm) are effective for *in vivo* imaging [33,35]. The proteins can be expressed in living cells and optical spectral separation techniques are becoming more developed and more widespread in the near-infrared spectrum [29,36,37].

This review describes challenges with bioluminescent imaging and advances in *in vivo* fluorescent imaging for systemic and deep tissue fungal infections, as well as discussing challenges and opportunities for widespread implementation of these technologies.

## 3. Key Challenges of *In Vivo* Imaging of Fungal Infections

There are a number of challenges to consider with *in vivo* imaging of pathogenic fungi. The recurrent problem of signal-to-noise is very important, especially with deep-seated infections. The presence of fatty tissue makes imaging more difficult as the excitation light must pass through the different barriers to excite the target. The nature of the infections and the different host infection niches have an important impact on the *in vivo* imaging strategies. Niches, such as the lung or brain, present additional technical challenges due to the presence of the rib cage, bones and cartilage, which reduce the transmission of light to the organ of interest. While localised infections can be visualised due to the high concentration of reporter-tagged organisms, systemic infections can present the additional challenge of a diffuse signal, requiring a brighter reporter to obtain a sufficient signal to monitor infection [27,38]. 

In addition, reporters for *in vivo* imaging of systemic fungal infections require specific optimisation, taking into consideration the different properties of fungi. For example, different growth forms of some fungi have to be taken into account. The morphological yeast–hyphal switch of certain pathogens, such as *C. albicans,* can represent a problem for quantification of the number of fungi present, as it is difficult to quantify the number of cells in a mass of hyphae. The change in morphology also represents a challenge for the construction of genetically encoded reporters as the promoter driving expression may only be expressed in one growth form, e.g., a synthetic luciferase was expressed in yeast but not in hyphae [39]. Another optimisation required for these reporters in some fungi is codon optimisation. *C. albicans,* for example, has an alternative codon usage where CTG codons, normally translated as leucine, are decoded as serine [40,41,42]. It is, therefore, necessary to change every CTG codon to an appropriate leucine codon to accurately translate the reporter protein. Cormack et al. [43] optimised a fully synthetic yeast-enhanced GFP gene (yEGFP3) for expression in *C. albicans* based on codons frequently used by this species. This molecular optimization is required for the construction and optimisation of any new reporter in *C. albicans.* However, lack of genome information for less-studied pathogens, such as *C. auris*, represent another challenge for the expansion of the use of fluorescent reporters *in vivo*. 

## 4. Bioluminescent Imaging

Bioluminescent imaging relies on the detection of visible light, known as bioluminescence, which arises from enzymatic oxidation of organic substrates by luciferases. Bioluminescence is seen in living organisms, such as fireflies and luminous marine microorganisms [28]. Bioluminescent imaging has the advantage of being highly sensitive, with excellent signal-to-noise ratios, and being non-invasive and non-toxic for animals [31]. It requires construction of genetically-encoded reporters that express proteins in living cells [31]. 

Three different luciferase genes have been developed to investigate fungal infections in living animals. Firefly luciferase (fLUC) from *Photinus pyralis* [44] can be visualised across a spectrum from yellow-green (550 nm) to red (620 nm). fLUC requires luciferin as a substrate, which is converted into oxyluciferin in an ATP-dependent reaction [45]. In contrast, the sea pansy (*Renilla reniformis*) rLUC [46] and the copepod *Gaussia princeps* gLUC [47] luciferases both use coelenterazine as their substrate and emit light at 480 nm [32]. These fungal bioreporters were generated through strong synthetic selection to enhance light emission compared to their parent bioluminescent molecules. Selection of an appropriate promoter was also required to make these reporters capable of being expressed within different fungal morphologies, such as *C. albicans* yeast and hyphal forms, to enable better understanding of pathogen biology and virulence [32,38].

Luciferase reporters have been used *in vitro* for many years in investigation of gene regulation. However, recent technological advances have allowed for the development of new, brighter reporters for bioluminescence study of fungal infections [27,32,38]. One of the first luciferase-expressing fungal species optimized for *in vivo* studies was reported by Doyle et al. [39], who developed a *C. albicans* codon optimised fLUC reporter to visualise systemic and vulvovaginal candidiasis [39]. Despite promising preliminary data, a decrease in light output was observed when yeast cells switched to a hyphal morphology [39]. This limitation encountered by Doyle et al. [39] could have been due to the expressed luciferase being targeted to the cytoplasm of the fungal cell, but may also have been a consequence of restricted luciferin permeability into *C. albicans* hyphae during infection [39,48].

To circumvent these problems, Enjalbert et al [32] created a cell surface-anchored *Gaussia princeps* luciferase (gLUC) to ensure expression on the surface of both yeast and hyphal cells [32]. This optimised reporter (emits at 480 nm) successfully enabled sensitive detection of *C. albicans* during oropharyngeal and vulvovaginal candidiasis in mouse models [39,49,50], but was unsuitable for imaging deep-seated infection due to a low signal-to-noise ratio at this wavelength. Dorsaz et al. [38] using a cytoplasmic-expressed, codon-optimised red-shifted firefly luciferase in *C. albicans* detected bioluminescent emissions in oropharyngeal candidiasis and also in systemic infection [38]. Although this reporter improved the detection limits of *C. albicans* cells in infected mice, it required administration of substrate D-luciferin immediately prior to imaging (10 minute *in vivo*), but signals *in vivo* were detected in the kidneys during systemic infection and within the tongue during oropharyngeal infection [38]. Similar results were obtained by Jacobsen et al. [27] where a codon-optimized firefly luciferase (emits light at >600 nm) showed good correlation between light emission and fungal burdens in *C. albicans* systemic infection [27]. A similar luciferase, optimized for expression in *Candida glabrata*, was recently successfully used to demonstrate successful antifungal therapy of *C. glabrata* biofilms on subcutaneously implanted catheters in mice [51]. Firefly luciferase is, therefore, considered the reporter of choice as a dynamic and sensitive bioluminescent reporter, but its use still requires administration of D-luciferin, which has to get into the fungal cells.

## 5. Fluorescent Imaging and iRFP

New technologies based on florescent and luminescent imaging have rapidly advanced in the past decade [31]. Fluorescent reporters enable brighter visualisation and require shorter exposure times than bioluminescence. Moreover, use of fluorescent proteins removes the need for substrates or cofactors, allowing real-time analysis of whole systems using live imaging [28].

Fluorescent proteins are easily expressed in living cells. Genes encoding the fluorescent proteins can be inserted into the fungal genome under the control of relevant promoters, providing valuable information about the activation of transcriptional networks underlying pathogenesis in animal models, which is already widespread in models of bacterial–host interactions [52,53]. 

Genetic engineering of different spectra, by changing the excitation and emission wavelengths of the protein, enhanced brightness and overcame background noise, which was a key step in optimisation of the first GFP reporter from *Aequorea victoria* (jellyfish) [54]. GFP derivatives were developed to emit five main colours: cyan, green, yellow, orange and red. However, these light wavelengths are absorbed by host tissue and, therefore, these reporters were not ideal for *in vivo* and deep tissue studies [55,56].

Fluorescent proteins emitting in the red and orange spectral region were originally developed from the marine coral reef anemone *Discosoma striata* [57]. The first fluorescent protein to emit in the red spectrum was DsRed, with an excitation peak at 558 nm and an emission peak at 583 nm [58]. DsRed has been expressed in *Aspergillus* and has proven to be a good reporter *in vitro*, but its faint signal makes it difficult to work with *in vivo* [59,60]. In addition, there were problems associated with the use of this new fluorescent protein, including incomplete maturation of DsRed. This was found to be due to DsRed being an obligate tetramer, forming large protein aggregates in living cells [61]. Mutagenesis to overcome this problem led to a second generation of monomeric red fluorescent proteins that avoided the problems of protein aggregates and toxicity [62]. mPlum, a monomeric far-red fluorescent protein with emission at 649 nm, is one of these DsRed derivatives [63]. A second promising fluorescent protein, HcRed, was isolated from the coral reef anemone *Heteractis crispa* [64]. HcRed is a dimer with a peak emission at 625 nm. Synthetic HcRed was engineered to encode two tandem molecules to overcome intermolecular dimerisation, but efforts to create a monomeric version by mutagenesis have been unsuccessful [64]. 

## 6. Near-Infrared Fluorescent Protein

A promising technological advance in fluorescent reporters is based on work with a bacterial phytochrome [33]. This phytochrome requires a cofactor, biliverdin, rather than a substrate to produce fluorescence. Biliverdin is a bile pigment, abundant in the mammalian host as the initial intermediate in haem catabolism [29,31,33]. IFP1.4 is a synthetic, optimised reporter that encodes the chromophore-binding domain from the *Deinococcus radiodurans* phytochrome, with excitation and emission maxima of 684 and 708 nm [33]. IFP1.4 is well expressed in mammalian cells and mice and can be visualised with or without the addition of exogenous biliverdin [33]. Exogenous addition increased near-infrared fluorescence, suggesting opportunities for further optimisation [33]. Subsequent mutagenesis generated a new, brighter near-infrared fluorescent protein (iRFP) with excitation at 690 nm and emission at 713 nm [33]. Despite innovative results with iRFP to visualise cells *in vivo*, the biliverdin cofactor limits its use, as biliverdin is absent in the lung and brain, limiting use of this reporter for fungal infections found in these niches [35]. Moreover, in order to increase the signal brightness, the fluorescent protein needs to be expressed on the surface of the cells to increase the chance of binding with the cofactor. 

These challenges may be overcome by the development of new bright reporters in the near-infrared spectrum, which are capable of being activated when expressed within the cytoplasm of the fungal cell. Scherbakova and Verkhusha [65] developed a range of novel iRFP proteins that efficiently combine with endogenous biliverdin, fluorescing without the need for exogenous biliverdin. Among these new iRFPs, iRFP670 shows the best result for *in vivo* imaging, and has since been successfully used for *in vivo* imaging in a mucosal lung infection model with *C. albicans* and *Pseudomonas aeruginosa* [66]. Due to this, iRFP670 is the most promising reporter exploitable for *in vivo* imaging. 

Along with the development of fluorescent reporters, microscopy methods have also advanced for *in vivo* investigations. Two-photon microscopy [67] allowed study of host–pathogen interactions in real time and in three dimensions, with high resolution imaging up to a depth of 1 mm [68]. However, in the original use of this technique, mice had to be anaesthetised and the chest cavity opened to permit imaging of fungal infections, such as *Aspergillus fumigatus* [69,70]. Development of two-photon and multi-photon intravital microscopy [71,72,73], has provided insights into fungal trafficking and host–fungal interactions. This relatively new tool has directly imaged targeting of *C. neoformans* to the central nervous system and neutrophil removal of the same fungus from the brain vasculature *in vivo* [74,75]. Despite the high resolution results and fully developed protocols [71], this technique is limited to the areas exposed under a surgically-implanted imaging window and cannot yet be used to image infections in the entire host [72]. As optical technologies and equipment become better developed and more widespread, these new reporters will supplement the existing palette of fluorescent proteins, focusing mainly on far red and near infrared reporters. This will allow further development of high resolution, full body *in vivo* imaging of fungal infections, allowing researchers to observe infection progression, or response to therapy, in individual animals.

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
