# Peer review of "A Bright Future for Fluorescence Imaging of Fungi in Living Hosts"

_jof, 2019, doi:10.3390/jof5020029_

Reviewer 1 Report

This is a review of fluorescent labeling techniques of fungi for use in live imaging. I am not aware of any reviews in this subject, and I found this one to be well written and succinct. This review will be a nice contribution to the literature.

Author Response

We thank the reviewer for their comments and for recognising the gap in the literature for this short review.

Reviewer 2 Report

Manuscript of Chapuis et al. describes different imaging techniques used to visualize infections caused by fungal pathogens. This review is nicely written and encounters with novel challenges during in vitro and in vivo imaging. Authors dedicate one paragraph to bioluminescent imaging. While reading this particular paragraph, I miss some findings dedicated to imaging of in vivo Candida albicans biofilm-related infections by bioluminescence. There are several publications regarding this topic. There is also a very recent publication dedicated to in vivo C. glabrata biofilm formation studied by bioluminescence. Therefore, I would suggest to consider my comment and include few sentences related to this topic.

Author Response

We thank the reviewer for their comments and for the suggestion to include a sentence on the use of a bioluminescence reporter to study biofilm formation.  We have now included this in the review and believe that this now gives a more complete picture of the use of this reporter in mycology studies. 

Editor's note: we also received a third opinion from an expert via email, please feel free to contact the editorial office if you would like to view.